# PasteGAN: A Semi-Parametric Method to Generate Image from Scene Graph

**Yikang Li**[1*], **Tao Ma**[2*], **Yeqi Bai**[3], **Nan Duan**[4], **Sining Wei**[4], **Xiaogang Wang**[1]

[1]The Chinese University of Hong Kong, [2]Northwestern Polytechnical University
[3]Nanyang Technological University, [4]Microsoft
`{ykli, xgwang}@ee.cuhk.edu.hk`, `taoma@mail.nwpu.edu.cn`
`baiyeqi@gmail.com`, `{sinwei, nanduan}@microsoft.com`

## Abstract

Despite some exciting progress on high-quality image generation from structured (scene graphs) or free-form (sentences) descriptions, most of them only guarantee the image-level semantical consistency, i.e. the generated image matching the semantic meaning of the description. They still lack the investigations on synthesizing the images in a more controllable way, like finely manipulating the visual appearance of every object. Therefore, to generate the images with preferred objects and rich interactions, we propose a semi-parametric method, PasteGAN, for generating the image from the scene graph and the image crops, where spatial arrangements of the objects and their pair-wise relationships are defined by the scene graph and the object appearances are determined by the given object crops. To enhance the interactions of the objects in the output, we design a Crop Refining Network and an Object-Image Fuser to embed the objects as well as their relationships into one map. Multiple losses work collaboratively to guarantee the generated images highly respecting the crops and complying with the scene graphs while maintaining excellent image quality. A crop selector is also proposed to pick the most-compatible crops from our external object tank by encoding the interactions around the objects in the scene graph if the crops are not provided. Evaluated on Visual Genome and COCO-Stuff dataset, our proposed method significantly outperforms the SOTA methods on Inception Score, Diversity Score and Fréchet Inception Distance. Extensive experiments also demonstrate our method's ability to generate complex and diverse images with given objects. The code is available at `https://github.com/yikang-li/PasteGAN`.

## 1 Introduction

Image generation from a scene description with multiple objects and complicated interactions between them is a frontier and pivotal task. With such algorithms, everyone can become an artist: you just need to define the objects and how they interact with each other, and then the machine will produce the image following your descriptions. However, it is a challenging problem as it requires the model to have a deep visual understanding of the objects as well as how they interact with each other.

There have been some excellent works on generating the images conditioned on the textual description [1, 2], semantic segmentations [3] and scene graphs [4]. Among these forms, scene graphs are powerful structured representations of the images that encode objects and their interactions. Nevertheless, nearly all the existing methods focus on the semantical compliance with the description on the image level but lack the object-level control. To truly paint the images in our mind, we need

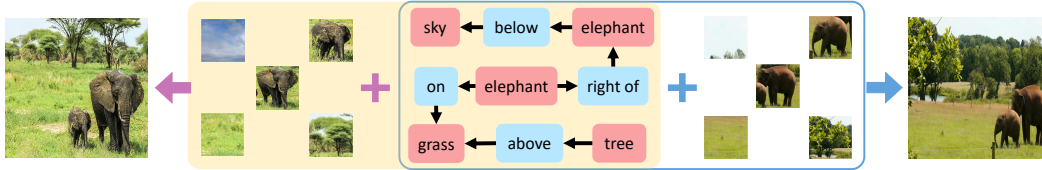

Figure 1: The image generation process is guided by both the scene graph and object crop images. Our proposed PasteGAN encodes the scene graph and given object crops, and generates the corresponding scene images. One group is in blue box and the other group is in the box with pale yellow background color. The appearance of the output scene image can be flexibly adjusted by the object crops.

to control the image generation process in a more fine-grained way, not only regulating the object categories and their interactions but also defining the appearance of every object.

Apart from the scene graph description, a corresponding anchor crop image is also provided for each defined object, which depicts how the object looks like. The synthesized image should follow the requirements: 1) the image as a whole should comply with the scene graph definition (denoted as *image-level matching*); 2) the objects should be the ones shown in the crop images (denoted as *object-level control*). Therefore, the original task is reformulated to a semi-parametric image generation from the scene graph, where the given object crops provide supervision on object-level appearance and the scene graph control the image-level arrangement.

In order to integrate the objects in the expected way defined by the scene graph as well as maintaining the visual appearance of the objects, we designed a Crop Refining Network as well as an attention-based Object-Image Fuser, which can encode the spatial arrangements and visual appearance of the objects as well as their pair-wise interactions into one scene canvas. Therefore, it can encode the complicated interactions between the objects. Then the Object-Image Fuser fuses all object integral features into a latent scene canvas, which is fed into an Image Decoder to generate the final output.

Sometimes, we just define the scene graph and don't want to specify the object appearance. To handle such situations, we introduce a Crop Selector to automatically select the most-compatible object crops from our external object tank. It is pre-trained on a scene graph dataset, which aims to learn to encode the entire scene graph and infer the visual appearance of the objects in it (termed as *visual codes*). Then the visual codes can be used to find the most matching object from the tank, where all the visual codes of external objects have been extracted offline using the scene graph they belong to.

Our main contributions can be summarized three folds: 1) we propose a semi-parametric method, PasteGAN, to generate realistic images from a scene graph, which uses the external object crops as anchors to guide the generation process; 2) to make the objects in crops appear on the final image in the expected way, a scene-graph-guided Crop Refining Network and an attention based Object-Image Fuser are also proposed to reconcile the isolated crops into an integrated image; 3) a Crop Selector is also introduced to automatically pick the most-compatible crops from our object tank by encoding the interactions around the objects in the scene graph.

Evaluated on Visual Genome and COCO-Stuff dataset, our proposed method significantly outperforms the SOTA methods both quantitatively (Inception Score, Diversity Score and Fréchet Inception Distance) and qualitatively (preference user study). In addition, extensive experiments also demonstrate our method's ability to generate complex and diverse images complying the definition given by scene graphs and object crops.

## 2   Related Works

**Generative models.** Generative models have been widely studied in recent years. Autoregressive approaches such as PixelRNN and PixelCNN [5] synthesize images pixel by pixel, based on the sequential distribution pattern of pixels. Variational Autoencoders [6, 7] jointly train an encoder that maps the input into a latent distribution and a decoder that generates images based on the latent distribution. In Generative Adversarial Networks (GANs) [8, 9], a pair of generator and discriminator are adversely optimized against each other to synthesize images and distinguish model synthesized images from the real image. In this work, we propose a novel framework using adversarial training strategy for image generation from scene graph and object crops.

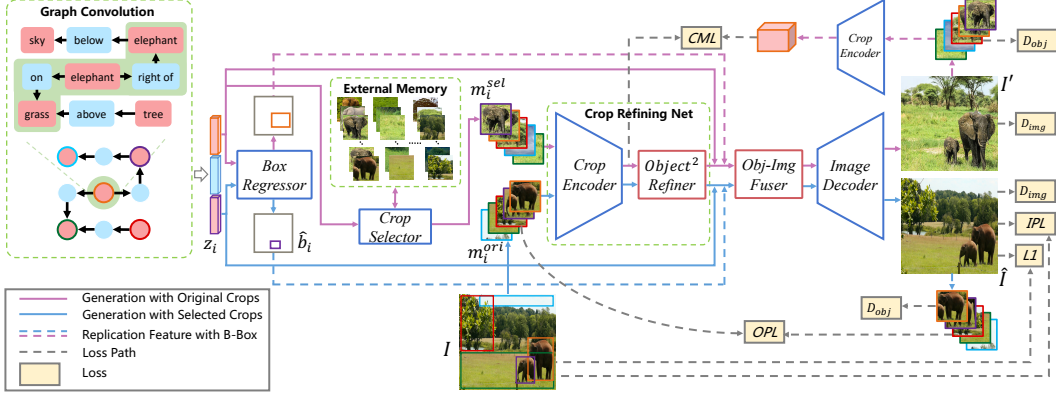

Figure 2: Overview of the training process of our proposed PasteGAN. The two branches are trained simultaneously with the same scene graph: the top branch focuses on generating the diverse images with the crops retrieved from the external memory, the bottom branch aims at reconstructing the ground-truth image using the original crops. The model is trained adversarially against a pair of discriminators and a number of objectives. L1, CML, IPL and OPL mean image reconstruction loss, crop matching loss, image perceptual loss and object perceptual loss respectively.

**Conditional Image Synthesis.** Conditional image synthesis aims to generate images according to additional information. Such information could be of different forms, like image labels [10, 11, 12], textual descriptions [1, 2, 13], scene graphs [4] and semantic segmentations [14, 3]. To synthesize photographic image based on semantic layout, Chen and Koltun [14] train a Cascaded Refinement Network (CRN) by minimizing Perceptual Loss [15, 16], which measures the Euclidean distance between the encoded features of a real image and a synthesized image. Qi *et al.* [3] extend this approach by filling segments from other training images into a raw canvas and refine the canvas into a photographic image with a similar approach. Different from the previous canvas building by simply stacking or averaging the objects [3, 4, 17], our integration process is implemented with a learnable 2D graph convolution architecture.

**Scene Graphs.** Scene graphs are directed graphs which represent objects in a scene as nodes and relationships between objects as edges. Scene graphs have been employed for various tasks, e.g., image retrieval [18], image captioning evaluation [19], sentence-scene graph translation [20], and image-based scene graph prediction [21, 22, 23]. Li *et al.* [24] proposes a convolutional structure guided by visual phrases in scene graphs. Visual Genome [25] is a dataset widely used by works on scene graphs, where each image is associated with a human-annotated scene graph.

Most closely related to our work, sg2im [4] generates image based on scene graph, graph convolution network [26] is employed to process scene graph information into object latent vectors and image layouts, which is then refined with a CRN [15] to optimize GAN losses [8, 9, 12] and pixel loss between synthesized and ground-truth images. Hong *et al.* [13] uses the textual information to generate object masks and heuristically aggregates them to a soft semantic map for the image decoding. Also, Zhao *et al.* [17] formulate image generation from layout as a task whose input is bounding boxes and categories of objects in an image. We further extend previous works by innovating the PasteGAN pipeline, where object crops are fused into images tractably; canvas generated from scene graph and object crops serves as enriched information and arms generated images with outstanding tractability and diversity.

## 3 The PasteGAN

The overall pipeline of our proposed PasteGAN is illustrated in Figure 2. Given a scene graph and selected object crops, our model generates a realistic image respecting the scene graph and the appearance of selected object crops. The training process involves two branches, one aims to reconstruct the ground-truth image using the original crops $m_i^{ori}$ (the bottom branch), the other focuses on generating the diversified images with the crops $m_i^{sel}$ retrieved from the external memory tank $\mathcal{M}$ (the top branch). The scene graph is firstly processed with Graph Convolution Network to get

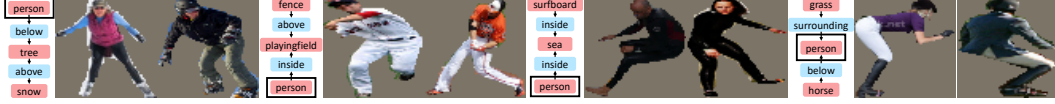

Figure 3: Top-2 retrieved ***person*** crops using our Crop Selector with different scene graphs during inference on COCO-Stuff Dataset.

a latent vector $z$ containing the context information for each object which is used for regressing the object location $\hat{b}_i$ and retrieving the most context-matching object crop by Crop Selector. Then Crop Encoder processes the object crops to encode their visual appearances. Afterwards, the crop feature maps and predicate vectors are fed into Object$^2$ Refiner to incorporate the pair-wise relationships into the visual features $v$. Object-Image Fuser takes the refined object feature maps concatenated with the expanded latent vectors and predicted bounding boxes as inputs to generate a latent scene canvas $L$. Similarly, the other latent scene canvas $\hat{L}$ is constructed along the reconstruction path using $m_i^{ori}$. Finally, Image Decoder reconstructs the ground-truth image $\hat{I}$ and generates a new image $I'$ based on $\hat{L}$ and $L$ respectively. The model is trained adversarially end-to-end with a pair of discriminators $D_{img}$ and $D_{obj}$.

**Scene Graphs.** Given a set of object categories $\mathcal{C}$ and a set of relationship categories $\mathcal{R}$, a scene graph can be defined as a tuple $(O, E)$, where $O = \{o_1, ..., o_n\}$ is a set of objects with each $o_i \in \mathcal{C}$, and $E \in O \times \mathcal{R} \times O$ is a set of directed edges of the form $(o_i, p_j, o_k)$ where $o_i, o_k \in O$ and $p_j \in \mathcal{R}$.

**External Memory Tank.** The external memory tank $\mathcal{M}$, playing a role of source materials for image generation, is a set of object crop images $\left\{ m_i \in \mathbb{R}^{3 \times \frac{H}{2} \times \frac{W}{2}} \right\}$. In this work, the object crops are extracted by the ground-truth bounding box from the training dataset. Additionally, once the training of our model has been finished, if the users do not want to specify the object appearance, $\mathcal{M}$ will provide the most-compatible object crops for inference. Note that the object crops on COCO-Stuff dataset are segmented by the ground-truth masks. The number of object crops on Visual Genome and COCO-Stuff dataset is shown in Table 1.

### 3.1 Graph Convolution Network

Following [4], we use a graph convolution network composed of several graph convolution layers to process scene graphs. Additionally, we extend this approach on 2-D feature maps to achieve message propagation among feature maps along edges while maintaining the spatial information. Specifically, feature maps $v_i \in \mathbb{R}^{D_{in} \times w \times h}$ for all object crops and predicate feature maps $v_p$ expanded from predicate vectors $z_p$ are concatenated as a triple $\left( v_{o_i}, v_{p_j}, v_{o_k} \right)$ and fed into three functions $g_s, g_p$, and $g_o$. We compute $v'_{o_i}, v'_{p_j} v'_{o_k} \in \mathbb{R}^{D_{out} \times w \times h}$ as new feature maps for the subject $o_i$, predicate $p_j$, and object $o_k$ respectively. $g_s, g_p$, and $g_o$ are implemented with convolutional networks.

The process of updating object feature maps exists more complexities, because an object may participate in many relationships and there is a large probability that these objects overlap. To this end, the edges connected to $o_i$ can be divided into two folds: the subject edges starting at $o_i$ and the object edges terminating at $o_i$. Correspondingly, we have two sets of candidate features $V_i^s$ and $V_i^o$,

$$V_i^s = \left\{ g \left( v_{o_i}, v_{p_j}, v_{o_k} \right) : (o_i, p_j, o_k) \in E \right\} \text{ and } V_i^o = \left\{ g \left( v_{o_k}, v_{p_j}, v_{o_i} \right) : (o_k, p_j, o_i) \in E \right\} \quad (1)$$

The output feature maps for object $o_i$ is then computed as $v'_{o_i} = h \left( V_i^s \cup V_i^o \right)$ where $h$ is a symmetric function which pools an input set of feature maps to a single output feature map (we use *average pooling* in our experiments). An example computational graph of a single graph convolution layer for feature maps is shown in Figure 4.

### 3.2 Crop Selector

Selecting a good crop for object $o_i$ is very crucial for generating a great image. A *good crop* is not simply matching the category, but also of the similar scene. Thus, to retrieve the *good crop*, we should also consider the context information of the object, i.e. the entire scene graph it belongs to.

Compared to hard-matching the edges connected to $o_i$, pre-trained sg2im [4] provides us a learning-based method to achieve this goal. It adopts a graph convolution network (GCN) to process scene graphs and a decoder to generate images. In order to generate the expected images, the output feature

of GCN should encode the object visual appearance as well as its context information (how to interact with other objects). So, after deprecating the image decoder part, the remaining GCN can be utilized as the Crop Selector to incorporating the context information into the visual code of $o_i$.

For each object $o_i$, we can compute its visual code by encoding its scene graph, and select the most matching crops in $\mathcal{M}$ based on some similarity metric (e.g. L2 norm). The visual codes of the object crops in external memory tank can be extracted offlinely. In our experiments, we randomly sample the crop from top-k matching ones to improve the model robustness and the image diversity. As shown in Figure 3, by encoding the visual code, our Crop Selector can retrieve different *person*s with different poses and uniforms for different scene graphs. These scene-compatible crops will simplify the generation process and significantly improve the image quality over the random selection.

### 3.3 Crop Refining Network

As shown is Figure 2, the crop refining network is composed of Crop Encoder and Object$^2$ Refiner.

**Crop encoder.** Crop encoder, aiming to extract the main visual features of object crops, takes as input a selected object crop $m_i \in \mathbb{R}^{3 \times \frac{H}{2} \times \frac{W}{2}}$ and output a feature map $v_i \in \mathbb{R}^{D \times w \times h}$. The object crops selected from external memory $\mathcal{M}$ are passed to into several 3×3 convolutional layers followed by batch normalization and ReLU layers instead of last convolutional layer.

**Object$^2$ Refiner.** Object$^2$ Refiner, consisting of two 2-D graph convolution layers, fuses the visual appearance of a series of object crops which are connected with relationships defined in the scene graph. As shown in Figure 4, for a single layer, we expand the dimension of predicate vector $z_{p_j}$ to the dimension of $D \times h \times w$ to get a predicate feature map $v_{p_j}$. Then, a tuple of feature maps $(v_{o_i}, v_{p_j}, v_{o_k})$ is fed into $g_s, g_p, g_o$, finally $avg$ averages the information and a new tuple of feature maps $(v'_{o_i}, v'_{p_j}, v'_{o_k})$ is produced. The new object crop feature map encodes the visual appearance of both itself and others, and contains the relationship information as well.

### 3.4 Object-Image Fuser

Object-Image Fuser focuses on fusing all the object crops into a latent scene canvas $L$. As we have faked an $'image'$ object, which is connected to every object through $'in\_image'$, we further extend the GCN to pass the objects to $'image'$. Firstly, we concatenate expanded object feature vector $z_{o_i}$ and object crop feature map $v_{o_i}$ to get an integral latent feature representation and replicate it with corresponding bounding boxes $\hat{b}_i$ to get $u_i \in \mathbb{R}^{D \times w \times h}$. Similarly, the new predicate feature map $u_{p_i}$ are concatenated through $v_{p_i}$ and expanded predicate vector $z_{p_i}$. Next, we select $u_{p_i}$ corresponding to $'in\_image'$ relationship and $u_i$ to calculate the attention map, where $f(\cdot) = W_f u$, $q(\cdot) = W_q u_p$,

$$\beta_i = \frac{\exp(t_i)}{\sum_{j=1}^{N} \exp(t_j)}, \text{ where } t_i = f(u_i)^T q(u_{p_i}), \tag{2}$$

and $\beta_i$ indicates the extent to which the fuser attends the $i^{th}$ object at every pixel. Then the output of the attention module is $u^{attn} = (u_1^{attn}, u_2^{attn}, ..., u_j^{attn}, ..., u_N^{attn})$, where

$$u^{attn} = \sum_{i=1}^{N} \beta_i l(u_i), \text{ where } l(u_i) = W_l u_i. \tag{3}$$

Finally, the last operation layer $h_{sum}$ aggregates all the object features into the $'image'$ object feature map by

$$y = \lambda_{attn} u^{attn} + u_{img}, \tag{4}$$

where $\lambda_{attn}$ is a balancing parameter. The latent canvas $L$ is formed by upsampling $y$ to $D \times H \times W$.

### 3.5 Image Decoder

The image decoder, based on a Cascaded Refinement Network (CRN), takes as input the latent scene canvas $L$ and generates an image $I'$ that respects the object positions given in $L$. A CRN consists of a series of cascaded refinement modules, with spatial resolution doubling between consecutive modules. The input to each refinement module is a channelwise concatenation of the latent scene canvas $L$ (downsampled to the input resolution of the module) and the feature map output by the previous refinement module. The input is processed by a pair of 3×3 convolution layers followed by batch

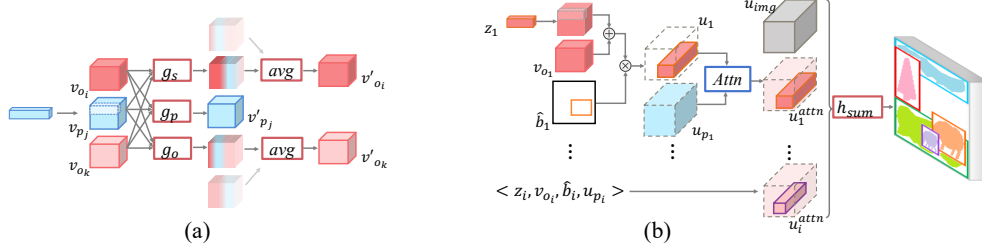

(a)                                    (b)

Figure 4: **(a) A single layer of Object$^2$ Refiner.** A tuple of feature maps $\left(v_{o_i}, v_{p_j}, v_{o_k}\right)$ is fed into $g_s, g_p, g_o$ for better visual appearance fusion. The last operation $avg$ averages all the information to form a new crop feature map. **(b) Object-Image Fuser.** Object latent vector $z_i$ is concatenated (symbol $\oplus$) with corresponding object crop feature map $v_i$ to form an object integral representation. Then the feature map is reproduced by filling the region (symbol $\otimes$) within the object bounding box $\hat{b}_i$ to get $u_i$, the rest of feature map are all zeros. Next, *Attn* takes as input $u_i$ and $'in\_image'$ predicate feature map $u_{p_i}$ to calculate the attention and form a new feature map $u_i^{attn}$. Finally, $h_{sum}$ sums all feature maps $u_i^{attn}$ to $'image'$ object feature map $u_{img}$ to form the latent scene canvas.

normalization and ReLU; the output feature map is upsampled using nearest-neighbor interpolation before being passed to the next module. The first refinement module takes Gaussian noise as input for the purpose of increasing diversity, and the output from the final module is processed with two final convolution layers to produce the output image $I'$.

## 3.6 Discriminators

We adopt a pair of discriminator $D_{img}$ and $D_{obj}$ to generate realistic images and recognizable objects by training the generator network adversarially. The discriminator $D$ tries to classify the input $x$ as real or fake by maximizing the objective

$$\mathcal{L}_{GAN} = \mathop{\mathbb{E}}_{x \sim p_{\text{real}}} \log D(x) + \mathop{\mathbb{E}}_{x \sim p_{\text{fake}}} \log D(1 - D(x)) \tag{5}$$

where $x \sim p_{\text{real}}$ represents the real images and $x \sim p_{\text{fake}}$ represents the generated images. Meanwhile, the generator network is optimized to fool the discriminators by minimizing $\mathcal{L}_{GAN}$.

$D_{img}$ plays a role of promoting the images to be realistic through classifying the input images, real images $I$, reconstructed ones $\hat{I}$ and generated ones $I'$, as real or fake. $D_{obj}$ takes as input the resized object crops cropped from real images and generated ones, and encourages that each object in the generated images appears realistic and clear. In addition, we also add an auxiliary object classifier which predicts the category of the object to ensure that the objects are recognizable.

## 3.7 Training

We end-to-end train the generator and the two discriminators $D_{img}$ and $D_{obj}$ in an adversarial manner. The generator is trained to minimize the weighted sum of eight losses:

**Image Reconstruction Loss.** $\mathcal{L}_1^{img} = \|I - \hat{I}\|_1$ penalizes the $L1$ differences between the ground-truth image $I$ and the reconstructed image $\hat{I}$, which is helpful for stable convergence of training.

**Crop Matching Loss.** For the purpose of respecting the object crops' appearance, $\mathcal{L}_1^{latent} = \sum_{i=1}^{n} \|v_i - v'_i\|_1$ penalizes the $L1$ difference between the object crop feature map and the feature map of object re-extracted from the generated images.

**Adversarial Loss.** Image adversarial loss $\mathcal{L}_{GAN}^{img}$ from $D_{img}$ and object adversarial loss $\mathcal{L}_{GAN}^{obj}$ from $D_{obj}$ encourage generated image patches and objects to appear realistic respectively.

**Auxiliary Classifier Loss.** $\mathcal{L}_{AC}^{obj}$ ensures generated objects to be recognizable and classified by $D_{obj}$.

**Perceptual Loss.** Image perceptual loss $\mathcal{L}_P^{img}$ penalizes the $L1$ difference in the global feature space between the ground-truth image $I$ and the reconstructed image $\hat{I}$, while object perceptual loss $\mathcal{L}_P^{obj}$ penalizes that between the original crop and the object crop re-extracted from $\hat{I}$. Mittal *et al.* [27]

| Dataset | COCO | VG |
|---------|------|-----|
| Train | 74 121 | 62 565 |
| Val. | 1 024 | 5 506 |
| Test | 2 048 | 5 088 |
| # Obj. | 171 | 178 |
| # Crops | 411 682 | 606 319 |

Table 1: Statistics of COCO-Stuff and VG dataset. # Obj. denotes the number of object categories. # Crops denotes the number of crops in the external memory.

| Method | IS ↑ | FID ↓ |
|--------|------|-------|
| Real Images | $16.3 \pm 0.4$ | - |
| w/o *Crop Selection* | $7.1 \pm 0.3$ | 96.75 |
| w/o *Object² Refiner* | $8.3 \pm 0.3$ | 61.28 |
| w/o *Obj-Img Fuser* | $8.7 \pm 0.2$ | 56.14 |
| full model | $9.1 \pm 0.2$ | 50.94 |
| full model (GT) | $10.2 \pm 0.2$ | 38.29 |

Table 2: Ablation Study using Inception Score (IS) and Fréchet Inception Distance (FID) on COCO-Stuff dataset.

used perceptual loss for images generated in the intermediate steps to enforce the images to be perceptually similar to the ground truth final image. So we add a light weight perceptual loss between generated images and ground-truth images to keep the perceptual similarity. And we found this loss would help to balance the training process.

**Box Regression Loss.** $\mathcal{L}_{box} = \sum_{i=1}^{n} \|b_i - \hat{b}_i\|$ penalizes the $L1$ difference between ground-truth and predicted boxes.

Therefore, the final loss function of our model is defined as:

$$\mathcal{L} = \lambda_1 \mathcal{L}_1^{img} + \lambda_2 \mathcal{L}_1^{latent} + \lambda_3 \mathcal{L}_{GAN}^{img} + \lambda_4 \mathcal{L}_{GAN}^{obj} + \lambda_5 \mathcal{L}_{AC}^{obj} + \lambda_6 \mathcal{L}_P^{img} + \lambda_7 \mathcal{L}_P^{obj} + \lambda_8 \mathcal{L}_{box} \quad (6)$$

where, $\lambda_i$ are the parameters balancing losses.

# 4 Experiments

We trained our model to generate $64 \times 64$ images, as an comparison to previous works on scene image generation [4, 17]. Apart from the substantial improvements to Inception Score, Diversity Score and Fréchet Inception Distance, we aim to show that images generated by our model not only respect the relationships provided by the scene graph, but also high respect the original appearance of the object crops.

## 4.1 Experiment Settings

**Datasets.** COCO-Stuff [28] and Visual Genome [25] are two datasets used by previous scene image generation models [4, 17]. We apply the preprocessing and data splitting strategy used by [4], the version of COCO-Stuff annotations is 2017 latest. Table 1 displays the attributes of the datasets.

**Implementation Details.** Scene graphs are argumented with a special $'image'$ object, and special $'in\_image'$ relationships connecting each true object with the *image* object; ReLU is applied for graph convolution and CRN; discriminators use LeakyReLU activation and batch normalization. The image and crop size are set to $64 \times 64$ and $32 \times 32$ correspondingly. We train all models using Adam [29] with learning rate 5e-4 and batch size of 32 for 200,000 iterations; training takes about $3 \sim 4$ days on a single Tesla Titan X. The $\lambda_1 \sim \lambda_8$ are set to 1, 10, 1, 1, 1, 1, 0.5 and 10 respectively. More details and qualitative results for ablation comparison can be found in the supplementary material.

## 4.2 Evaluation Metrics

**Inception Score.** Inception Score [30] computes the quality and diversity of the synthesized images. Same as previous work, we employed Inception V3 [31] to compute Inception Score.

**Diversity Score.** Different from the Inception Score that calculates the diversity of the entire set of generated images, Diversity Score measures the perceptual difference between a pair of images. Same as the former approach of calculating Diversity Score [17], we use the Alex-lin metric [32], which inputs a pair of images into an AlexNet and compute the $L2$ distance between their scaled activations.

**Fréchet Inception Distance.** FID is a more robust measure because it penalizes lack of variety but also rewards IS, and is analogous to human qualitative evaluation.

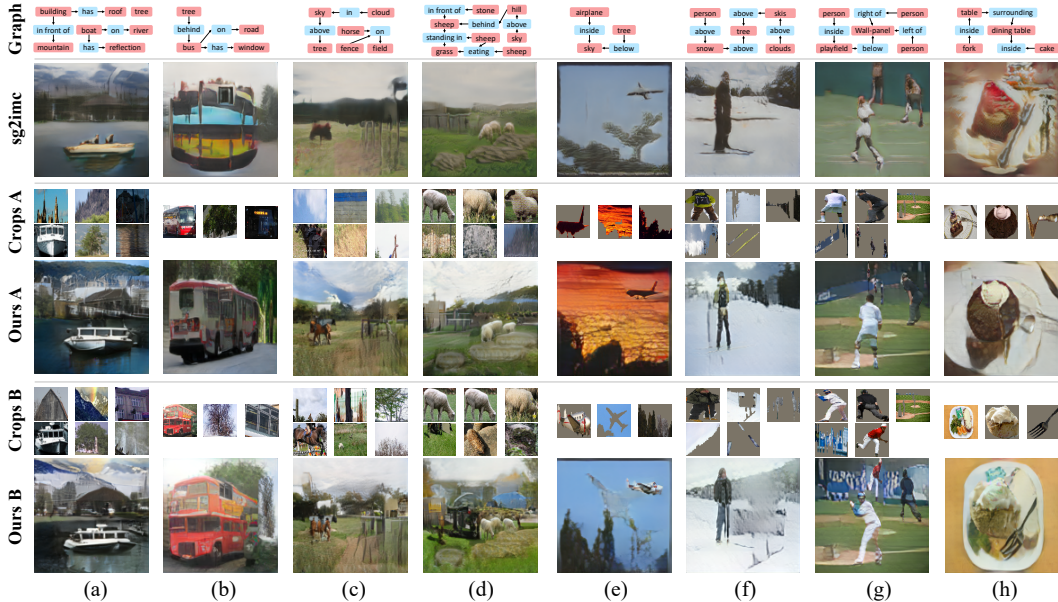

Figure 5: Examples of $64 \times 64$ generated images using sg2im and our proposed PasteGAN on the test sets of VG (left 4 columns) and COCO (right 4 columns). For each example we show the input scene graph and object crops selected by Crop Selector. Some scene graphs have deprecated similar relationships and we show at most 6 object crops here. Please zoom in to see the details between crops and generated images.

## 4.3 Comparison with Existing Methods

Two state-of-the-art scene image generation models are compared with our work. **sg2im** [4]: Most related to our work, we take the code[2] released by sg2im to train a model for evaluation. **layout2im** [17]: We list the Inception Score and Diversity Score reported by layout2im. For sake of fairness, we provide our model with ground truth boxes in comparison with layout2im. Table 3 shows the performance of our model compared to the SOTA methods and real images.

| Method | Inception Score ↑ | | Diversity Score ↑ | | FID ↓ | |
|---|---|---|---|---|---|---|
| | COCO | VG | COCO | VG | COCO | VG |
| Real Imgs | $16.3 \pm 0.4$ | $13.9 \pm 0.5$ | - | - | - | - |
| sg2im | $6.7 \pm 0.1$ | $5.5 \pm 0.1$ | $0.02 \pm 0.01$ | $0.12 \pm 0.06$ | 82.75 | 71.27 |
| PasteGAN | $\mathbf{9.1 \pm 0.2}$ | $\mathbf{6.9 \pm 0.2}$ | $\mathbf{0.27 \pm 0.11}$ | $\mathbf{0.24 \pm 0.09}$ | **50.94** | **58.53** |
| sg2im (GT) | $7.3 \pm 0.1$ | $6.3 \pm 0.2$ | $0.02 \pm 0.01$ | $0.15 \pm 0.12$ | 63.28 | 52.96 |
| layout2im | $9.1 \pm 0.1$ | $8.1 \pm 0.1$ | $0.15 \pm 0.06$ | $0.17 \pm 0.09$ | - | - |
| PasteGAN (GT) | $\mathbf{10.2 \pm 0.2}$ | $\mathbf{8.2 \pm 0.2}$ | $\mathbf{0.32 \pm 0.09}$ | $\mathbf{0.29 \pm 0.08}$ | **38.29** | **35.25** |

Table 3: Performance on COCO-Stuff and VG datsaset in Inception Score, Diversity Score and Fréchet Inception Distance (FID).

## 4.4 Qualitative Results

We use sg2im model released to generate images for comparison with ours. Each scene graph is paired with two sets of object crops for our PasteGAN. Figure 5 shows example scene graphs from the VG and COCO-Stuff test sets and corresponding generated images using sg2im and our method.

Both two methods can generate scenes with multiple objects, and respect the relationships in the scene graph; for example in all the three images in Figure 5 (a) we see a boat *on* the river, which *has* its own reflection. More importantly, these results indicate that with our method, the appearance of the output scene image can be flexibly adjusted by the object crops. In (a), the ship in crop set A has a white bow and the ship in crop set B has a black bow, and this is highly respected in our generated images. Similarly, as shown in (b), our generated image A contains a bus in white and red while generated image B contains a bus in red; this respects the color of the bus crops A and B.

It is clear that our model achieves much better diversity. For example in (d), the sheeps generated by sg2im looks almost the same, however, our model generates four distinct ships with different colors and appearances. This is because sg2im forces their model to learn a more general representation of the *sheep* object, and stores the learned information in a single and fixed word embedding. Additionally, the selected crops also clearly proves Crop Selector's powerful ability of capturing and utilizing the information provided by scene graphs and object crops' visual appearance. Our model represent an object with both the word embedding and the object crop, this provides PasteGAN the flexibility to generate image according to the input crops.

### 4.5 Ablation Study

We demonstrate the necessity of all components of our model by comparing the image quality of several ablated versions of our model, shown in table 2. We measure the images with Inception Score and Fréchet Inception Distance. The following ablations of our model is tested:

**No Crop Selection** omits Crop Selector and makes our model utilize random crops from same categories. This significantly hurts the Inception Score, as well as FID, and visible qualities, since irrelevant or unsuitable object crops are forced to be merged into scene image, which confuses the image generation model. Diversity Score doesn't decrease because the random crops preserve the complexity accidentally. The results in supplementary material further demonstrate the powerful ability of Crop Selector to capture the comprehensive representation of scene graphs for selecting crops.

**No Object$^2$ Refiner** omits the graph convolution network for feature map fusing, which makes the model fail to utilize the relationship between objects to better fuse the visual appearance in the image generation. That Inception Score decreases to 8.3 and FID becomes 61.28 indicate using the feature maps of crops straightly without fusion is harmful to the generation. It tends to produce overrigid and strange images.

**No Object-Image Fuser** omits the last fusion module, generating the latent canvas only by replicating features within bounding boxes. We observe worse results on both the visual quality and the quantitative metrics. More qualitative results are shown in supplementary material.

## 5 Conclusion

In this paper we have introduced a novel method for semi-parametrically generating images from scene graphs and object crops. Compared to leading scene image generation algorithm which generate image from scene graph, our method parametrically controls the appearance of the objects in the image, while maintaining a high image quality. Qualitative results, quantitive results, comparison to a strong baseline and ablation study justify the performance of our method.

### Acknowledgments

This work is supported in part by SenseTime Group Limited, in part by Microsoft Research, and in part by the General Research Fund through the Research Grants Council of Hong Kong under Grants CUHK14202217, CUHK14203118, CUHK14207319.

## Footnotes

*Equally contributed to the work.

[2]https://github.com/google/sg2im

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
