[Supplementary Material · supplementary_material.pdf]

# Submission 2179 Supplementary Material

## 1 User Studies

Inception score and diversity score are convenient automatic evaluation metrics that provide coarse measures of image quality; another important measure is human judgement of the generated images. Therefore, two user studies are performed on Mechanical Turk to evaluate our results.

We compare our method with sg2im, the SOTA method for generating image from scene graph. We compare $64 \times 64$ images generated by sg2im and our PasteGAN on COCO dataset's scene graphs. For user studies, five workers repeat all trails.

### 1.1 Scene Graph Matching

We measure semantic interpretability by showing users a COCO scene graph, and the corresponding images generated by sg2im and PasteGAN. We ask users to select the image that better matches the scene graph. Shown in Figure 1 are a an example image pair and the results. A majority of workers prefer image generated by our approach in 78.4% of the image pairs. This suggests that compared to sg2im our method more frequently generates semantically meaningful images that highly respect the input scene graphs.

*Which image matches the scene graph better?*

| | sg2im | Ours |
|---|---|---|
| **User Choice** | 216 / 1000 (21.6%) | **784** / 1000 **(78.4%)** |

Figure 1: We performed a user study to compare the semantic interpretability of our method against sg2im. Top: We use sg2im to generate an image from a COCO scene graph, and use our method to generate an image from the same scene graph. We show users the scene graph and both images, and ask which better matches the scene graph. Bottom: Across 1000 val image pairs, users prefer the results from our method by a large margin.

### 1.2 Objects Recall

Another important measure is the number of recognizable objects in the generated images. As conducted by sg2im, in each trial we show an image from one method and a list of COCO objects and ask users to identify which objects appear in the image. An example and results are shown in Figure 2. We compute the fraction of objects that a majority of users believed were present, dividing

*Which objects are presented?* Cloud, sky, tree, fence, field, horse

| | sg2im | Ours |
|---|---|---|
| **Thing Recall** | 847 / 1780 (47.6%) | **1047** / 1780 **(58.8%)** |
| **Stuff Recall** | 2341 / 3785 (61.8%) | **2984** / 3785 **(78.8%)** |

Figure 2: We performed a user study to measure the number of recognizable objects in images from our method and from sg2im. Top: We use sg2im to generate an image from a COCO scene graph, and use our method to generate an image from the same scene graph. For each image, we ask users which COCO objects they can see in the image. Bottom: Across 1200 val image pairs, we measure the fraction of things and stuff that users can recognize in images from each method. Our method produces more recognizable objects.

the results into things and stuff. Both methods achieve higher recall for stuff than things, and our method achieves significantly higher object recall, with 23.6% and 27.5% relative improvements for thing and stuff recall respectively.

## 1.3 Crop Matching

We propose this experiment to measure the similarity between objects in the generated images and their corresponding crops from the memory bank. Provided the object-crop pairs in each image, the user rates the similarity between the objects and crops into 1 to 5, where 1 stands for "not similar at all" and 5 stands for "highly similar". The final similarity scores is calculated as the average of similarity rating of all the object-crop pairs. An example and results are shown in Figure 3. Our model achieves 54.6% similarity score, which indicates that the generated images highly respect the original crops from the memory bank.

*How similar are each pair of crop and generated object to each other?* Rate from 1 to 5

| **Similarity Score** | **3148** / 5765 **(54.6%)** |
|---|---|

Figure 3: We performed a user study to measure the similarity between between objects in the generated images and their corresponding crops from the memory bank. Top: We use our PasteGAN to generate an image from a COCO scene graph, and store the utilized crops. For each image, we ask users to rate the similarity between the objects and crops into 1 to 5. Bottom: Across 1500 val image pairs, we sum the similarity rating and take their average as the similarity score.

## 2  Box Regressor

We predict bounding boxes for images using a box regressor. The input to the box regressor are the final embedding vectors $v'$ for objects produced by the graph convolution network. The output from the box regressor is a predicted bounding box for the object, parameterized as $(x_0, y_0, x_1, y_1)$ where $x_0$, $x_1$ are the left and right coordinates of the box and $y_0$, $y_1$ are the top and bottom coordinates of the box; all box coordinates are normalized to be in the range [0, 1].

## 3  Ablation Study

Extensive anlation experiments demonstrate our different modules' powerful ability to perform their own functions and cooperate with each other to generate complex and diverse images with given objects.

### 3.1  No Crop Selection.

No Crop Selection omits the crop selector and makes our model utilize random crops from same categories. We show the examples for comparison in Figure 4.

Figure 4: Irrelevant or unsuitable object crops are forced to be merged into scene image, which confuses the image generation process.

### 3.2  No Crop Refiner.

No Crop Refiner omits the graph convolution network for feature map fusing, which makes the model fail to utilize the relationship between objects to better fuse the visual appearance in the image generation. The results are shown in Figure 5.

Figure 5: That no crop refiner makes the model fail to utilize the relationship between objects to better fuse the visual appearance in the image generation. We can see the fuzzy objects in generated images, which indicate that our crop refiner is crucial for feature map refining.

### 3.3  No Object-Image Fuser.

No Object-Image Fuser omits the last fusion module, generating the latent canvas only by replicating features within bounding boxes. The comparison examples are shown in Figure 6.

Figure 6: No Object-Image Fuser. In top row images, the edges of objects are preserved directly without any fusion. In bottom row images, the objects show more interactions with others objects or surroundings. And these images look more realistic and vivid.

## 4  Samples

We show some extend results on Visual Genome dataset in Figure 7 and on COCO-Stuff dataset in Figure 8. Both Figure 7 and Figure 8 show our model's performance. And in Figure 7, the bottom 2 rows show typical cases compared to ground-truth images. The ground-truth image of penultimate row contains a plain while the two generated images both contain a lake. The ground-truth image of last row includes a plane but no planes are in the two generated images. However, our results already respect the scene graph and object crops.

Figure 7: Samples on Visual Genome dataset. The bottom 2 rows show typical cases compared to ground-truth images.

Figure 8: Samples on COCO-Stuff dataset. Please zoom in to see the details between object crops and images.