[Reviews · NeurIPS 2019]

Reviewer 1



- I think there are quite a few novel and interesting ideas in this work. The crop selection is not a trivial problem and the authors use the visual code of each object for matching. It looks interesting but I do not quite understand how well this approach works. I suggest the authors can make some visualization on this. - What are u_{img} and u^{attn}? What do they refer to? - The paper looks technically interesting but I am not if all the components are necessary. For example, is D_{obj} really useful? I think D_{img} is probably enough and D_{img} has more global information than objects. - How are the hyper-parameters tuned? The lambdas are supposed to be weights to balance different terms in the loss function. - Not sure if the code will be available.

Reviewer 2



2.1. Originality. The proposed method contains a lot of components. Some of them are well known already. In [16] a very similar architecture is proposed. The differences are that this work uses text, while [16] directly requires layout with bboxes and this one requires object crops, while the work in [16] generates objects from scratch. Overall, I think the originality of this method is only okay. Methodological novelty of the presented method, I believe, is marginal 2.2. Quality. The presented method seems to outperform both [16] and [4]. I would consider the numbers with caution, since the presented method uses ground truth image patches to generate images. Therefore, inception scores and diversity scores are much higher, especially the diversity score. Clearly, by sampling different patches you get these numbers higher, compared to works directly generating pixels. In section 4.4 the qualitative benefits of the presented method are discussed, while this simple advantage is omitted. Overall quality of the generated results is still far from realistic. Would it be possible to generate higher resolution samples, like 256x256? I don't think we should stick to 64x64 for comparison purposes. 2.3. The paper is well written, easy to follow. At least one reference/comparison is missing [Hong et al.]. Similarly to the presented work, they also first generate bounding boxes from text. They don't use any patches though. 2.4. Significance. The paper marginally moves the results in text-to-image synthesis forward. The paper proposes no groundbreaking contributions to the reader. Hong et al. Inferring Semantic Layout for Hierarchical Text-to-Image Synthesis, CVPR'2018.

Reviewer 3



Limited novelty: The proposed approach is closely related to two lines of related work: 1) sg2im [4] which generates images from scene graph representations, and 2) semi-parametric image synthesis [3], which leverages semantic layouts and training images to generate novel images. The key difference to sg2im is the use of image crops in order to perform semi-parametric synthesis; however, in comparison to prior work on semi-parametric methods [3], as suggested by the authors (Line 82-83) the primary difference is the use of graph convolution architecture, where a similar graph convolution method has been introduced in [4]. I’d like to see more justifications from the authors regarding the technical novelty of this approach in presence of these two lines of work. Limited resolution: My concern about the limited novelty is exacerbated by the fact that the generated images are still in low-resolution (64x64) as prior work [4], even though high-resolution image crops are used to aid the image generation process. In contrast, related work [3] is able to generate images of much higher resolutions, e.g., 512x1024, using their semi-parametric method (which was not compared in the experiment). Could the authors comment on the possibility of using this proposed method in generating high-resolution images? The experiment results could be much stronger if the authors can demonstrate the effectiveness of this method in generating larger images. Crop selection: It is unsatisfying that the crop selector relies on pretrained models from prior work [4] to rank crop candidates, instead of jointly learned with the rest of the model. Is there a way to make the crop selector training as part of the final learning objective? Relations of scene graphs: The model is trained adversarially with two discriminators on both object level and image level. However, there seems to be no training objective to ensure the pairwise relationships in the generated images to match the edges of the scene graphs. Is there any other learning objective that can ensure the consistency of the relationships between the scene graph and its corresponding generated image? Figure 2: Is there a mistake in the caption description, which is inconsistent with the main text? Does the top branch generate the new image while the bottom branch reconstructs the ground-truth?

[Author Response · NeurIPS 2019]



Figure A. Top-2 retrieved **person** crops using our crop selector with different scene graphs during inference on COCO.

**The performance of the proposed Crop Selection. (R1)** As shown in Fig. A, by encoding the context information
with GCN, our selector can retrieve different *person*s with different poses and uniforms for different scene graphs.
These scene-compatible crops will simplify the generation process and significantly improve the image quality over the
random selection. The ablation study *"w/o crop selection"* in Tab. 2 validates our claim.

**What do** $u_{img}$ **and** $u^{attn}$ **refer to. (R1)** Object-Image Fuser aims at fusing different objects to the virtual *"image"*
object via *"in image"* relationship. Objects are merged through attention maps. $u^{attn}$ is the fused object feature
after calculating attentions with *"in image"* relationships. $u_{img}$ is the feature map of virtual *"image"* object. Eq. (4)
aggregates the objects to the *"image"* via $\lambda_{attn}$ (we use 1 in our experiments). Additionally, there is a typo in Eq. (3),
where $D$ should be $N$, the number of objects in that image. Will revise in the next version.

**Necessity of components. (R1)** Our PasteGAN is based on SG2IM [4]. Compared to SG2IM, our method has several
novelties: 1) a semi-parametric setting; 2) an object-image fuser; 3) a crop selector; 4) an object$^2$ refiner. Our ablation
study (Tab. 2) shows the necessity of these components. For the other components like $D_{img}$ or $D_{obj}$, they are not our
main contributions, and their effectiveness has been validated in [4]. We got similar conclusions on our PasteGAN.

**Reproducibility of PasteGAN. (R1, R2)** Most of hyper-parameters are used in [4] and we followed their settings. For
the newly-introduced ones, the hyper parameters are tuned heuristically. We will release the code for reproducibility.

**Crop's resolution. (R1)** Comparing to the size of an image, most objects usually occupy a small region, so we specify
the size of an object to half of the image size to reduce the computational cost. Experimental results show that the
image-size crops don't bring significant improvements and slower the inference.

**Limited novelty. (R2, R3)** Our proposed PasteGAN aims at enabling the model to finely control the appearance of
the objects in generated image through a semi-parametric setting. As R2 said, the semi-parametric setting has its
advantages naturally over generating the images from scratch, and to the best of our knowledge, our PasteGAN is the
first semi-parametric method to generate the image from the scene graph. Moreover, this is not a trivial combination of
[3] and [4], as we propose: 1) a crop selector to retrieve the most scene-matching crops by encoding the scene graph; 2)
an object$^2$ refiner to translate the object crops to the target appearance based on their connections; 3) an object-image
fuser to merge the objects into a latent scene feature map. All these modules make our PasteGAN outperform the
baselines, which is proven by our experimental results. Additionally, the high diversity score and the qualitative results
both prove that by alternating the input object crops, we can finely control the generated images, especially the objects'
appearance, and synthesize diverse images, which is the goal of this work.

**Limited resolution. (R2, R3)** Most of the SOTA high-resolution generative models either leverage a cascaded generator,
like StackGAN++, or generate specific images, such as birds or street views, like [3]. As generating general images
from scene graphs is an emerging and challenging topic, how to encode the scene graph is the most critical part of
the investigation, so most of the existing methods choose to generate low-resolution images for more efficient model
training. For a fair comparison with baselines, SG2IM [4] and Layout2IM [16], we follow the most commonly-used
size, $64 \times 64$. We also evaluate our method on $128 \times 128$ image generation. We get **14.5** and **10.4** (Inception Score) on
COCO and VG, respectively, which outperforms 12.4 (COCO) of [Hong et al.] on the same size. For higher resolution,
like $256 \times 256$, we may need to redesign a multi-stage cascaded generator, which worths investigating in the future.

**Missing reference of [Hong et al.]. (R2)** The two methods work on different but related problems, image generation
from free-form texts v.s. from more structured scene graphs. Besides the semi-parametric setting, the most significant
difference of the methodology part is that [Hong et al.] uses the textual information to generate object masks and
heuristically aggregates them to a soft semantic map for the image decoding, while ours utilizes a 2D-GCN-based
*object-image fuser* to merge the objects in a learnable way. We will cite and discuss it in the next version.

**Crop Selection. (R3)** If the crop selector is jointly trained with our PasteGAN, the visual code of each object will
change at every training step. Correspondingly, we need to extract the visual code for all the training set whenever the
model gets updated, which is nearly impossible. Therefore, we alternatively utilize the pre-trained model to extract the
visual codes offline and directly use them during the training and inference.

**Pair-wise relationships discriminator. (R3)** Using an additional relationship discriminator to ensure the consistency
of the pair-wise relations is a great idea worth trying. We used a GCN-based RelationNet to distinguish the pair-wise
relations in the generated image as the additional discriminator, but cannot observe any improvements on these two
benchmark datasets. We will continue investigating the idea in the future.

**Mistake in captions of Fig. 2. (R3)** Will revise in the next version.

[Meta-Review · NeurIPS 2019]

This submission received borderline positive reviews. While the reviewers ultimately did not reach consensus during the discussion period, one did step forward to 'champion' the paper, and another was supportive of this decision. This submission is a 'systems paper,' and should be evaluated as such. The paper does not focus on new algorithmic results, but rather on building a nontrivial system to achieve impressive results on an important problem, and it justifies its design decisions (e.g. with an ablation study). There is some concern about the output images being low-resolution. However, the authors convincingly argue in their response that there exist techniques that could be adapted to their setting for increasing the output resolution, and that this should be viewed as orthogonal to their contribution. Finally, the paper *does* actually include a novel technical contribution, which is in its module for selecting appropriate object crops.